# Sensory Phenotype of the Oesophageal Mucosa in Gastro-Oesophageal Reflux Disease

**DOI:** 10.3390/ijms24032502

**Published:** 2023-01-28

**Authors:** Ahsen Ustaoglu, Philip Woodland

**Affiliations:** Wingate Institute of Neurogastroenterology, Blizard Institute, Faculty of Medicine and Dentistry, Queen Mary University of London, London E1 4NS, UK

**Keywords:** GORD, peripheral sensitization, heartburn, oesophageal mucosa

## Abstract

Gastroesophageal reflux disease (GORD) affects up to 20% of Western populations, yet sensory mechanisms underlying heartburn pathogenesis remain incompletely understood. While central mechanisms of heartburn perception have been established in earlier studies, recent studies have highlighted an important role of neurochemical, inflammatory, and cellular changes occurring in the oesophageal mucosa itself. The localization and neurochemical characterisation of sensory afferent nerve endings differ among GORD phenotypes, and could explain symptom heterogeneity among patients who are exposed to similar levels of reflux. Acid-induced stimulation of nociceptors on pain-sensing nerve endings can regulate afferent signal transmission. This review considers the role of peripheral mechanisms of sensitization in the amplification of oesophageal sensitivity in patients with GORD.

## 1. Introduction

The most troublesome symptom of gastroesophageal reflux disease (GORD), heartburn, is generated by the perception of noxious refluxate from the lumen by sensory pathways innervating the oesophageal epithelium. Pain under inflammatory conditions has been well established in disease states including erosive reflux disease (ERD), which is characterized by visible mucosal injury at endoscopy, and pain is a protective and cardinal feature of acute inflammation [1]. However, 60–70% of GORD patients with troublesome symptoms have no macroscopic injury (non-erosive reflux disease, NERD), and up to 30% of patients have incomplete symptom response to proton pump inhibitors (PPIs) [2,3]. Therefore, there is a need for better understanding of mechanisms underlying heartburn in the absence of overt inflammation. Increased sensitivity to gastroesophageal reflux has been highlighted in PPI-refractory NERD patients who demonstrated hypersensitivity to acid and, in some cases, saline perfusion, compared to their erosive counterparts and healthy controls [4,5,6]. Meanwhile, many Barrett’s oesophagus (BO) patients who have had years of pathological acid reflux often do not present GORD symptoms, yet functional heartburn (FH) patients perceive troublesome heartburn despite having no association with reflux events.

The wide discrepancy of symptoms observed between GORD patients with similar levels of acid exposure is important to consider in relation to the sensory characterization of the oesophageal mucosa. While the components of the refluxate can vary, and central mechanisms of sensitization such as psychological stress can affect symptom severity, the heterogeneity in symptom profiles across GORD may be contributed to by the neurochemical, genetic, and cellular differences of the oesophageal mucosa itself. In this review, we discuss the current understanding of the mucosal sensory phenotype of GORD.

## 2. Peripheral Sensitization in GORD

Increased sensitization of spinal dorsal horn neurons through repeated oesophageal mucosal injury during a process known as secondary hyperalgesia likely plays an important role in heartburn pathogenesis, as shown by studies investigating the oesophagus and brainstem neurons after acute infusion of acid and pepsin in animal models [7,8]. However, the magnitude of sensory response can be increased by excessive noxious stimulation and eventual tissue damage which triggers inflammatory mediator release, leading to a reduced transduction threshold of cation channels expressed on the peripheral terminals of specialized pain-sensing neurons named nociceptors [9]. The exposure of nociceptors to inflammatory mediators, including low pH, can thus be considered a common pathway that reduces the resting membrane potential at the peripheral nerve terminal through the leakage of cations, leading to action potential generation. The molecular mechanisms underlying this process of peripheral sensitization can involve the activation of G-protein-coupled receptors (GPCRs) on peripheral nociceptor terminals by inflammatory mediators such as histamine, bradykinin, or prostaglandin, in turn phosphorylating ion channels in an autocrine manner and reducing the activation threshold of the nociceptor (Figure 1A). Moreover, upregulation of the ion channels themselves as a response to neuropeptides such as nerve growth factor (NGF), and their subsequent translocation to the cell body and nerve terminals can further enhance sensitivity of the nerve (Figure 1B). Neuroimmune interactions, with the release of neuropeptides such as substance P (SP) or calcitonin gene-related peptide (CGRP) activating mast cells to release NGF, can further feed into the bidirectional cycle (Figure 1C) [10]. Sensitization occurs when a persistent noxious stimulus induces a combination of these peripheral intracellular mechanisms.

The induction of GORD symptoms is likely to be partly transduced by acid-sensing receptors expressed on nociceptors in the oesophageal mucosa. Transient receptor potential vanilloid type-1 (TRPV1), a non-selective calcium channel that is frequently involved in pain pathways and can be activated by acid, could be a sensory transductor of reflux-induced symptoms [11,12]. The nociceptive role of TRPV1 was highlighted in cat oesophageal mucosa, where its acid-induced activation stimulated intrinsic neuronal release of SP and CGRP neuropeptides which activate spinothalamic tract neurons and induce neurogenic inflammation [13]. Recent studies have demonstrated increased TRPV1 mRNA expression in GORD patients compared to healthy controls, and its localization on nerve endings expressing pan-neuronal marker protein gene product 9.5 (PGP 9.5) in NERD patients, further highlighting the dependence of its expression on increased acid exposure [14,15,16]. Importantly, a rat model of ERD demonstrated the blockade of acid-induced activation of vagal sensory afferent nerves when administered with TRPV1 antagonist AMG9810, suggesting a regulatory role for TRPV1 in peripheral sensitization in response to acid challenge [17]. Electrophysiological studies have highlighted that capsaicin injection into the marrow cavity causes increased sensory activity of C fibers and Aδ fibers, further supporting the role of TRPV1 in pain sensation [18].

Transient receptor potential melastatin 8 (TRPM8), a cold and menthol-induced sensory transductor that is most often expressed in a subgroup of TrkA^+^ sensory neurons, may be another candidate for sensory transduction in the oesophageal mucosa [19]. A recent in vivo study in humans demonstrated how oesophageal infusion of TRPM8 agonist menthol induced heartburn in GORD patients, while the infusion of cold water induced oesophageal pain [20]. Moreover, the sensory role of TRPM8 was further highlighted in guinea pig oesophagus, where TRPM8 expression was observed on jugular C fibers, and acid perfusion evoked action potentials in jugular C fibers, but not nodose C fibers [21]. Collectively, these studies suggest a distinctive nociceptive role for TRPM8 in heartburn pathogenesis.

Acid-sensing ion channels (ASICs) are sensors of extracellular pH which belong to a voltage-insensitive, amiloride-sensitive family of cation channels that are solely activated by extracellular protons, making them a prime candidate for sensation in heartburn patients [22]. Much like TRPV1, activation of ASICs leads to an influx of Na^+^, which in turn desensitizes the neuron and initiates an action potential, feeding into nociceptive pathways in the central nervous system. Recent findings of ASIC expression on visceral sensory neurons highlight ASICs as likely sensors of acid-induced pain [23]. One such study in mouse skin found ameliorated sensitivity to noxious pinch and acid infusion in ASIC3 knockout mice compared to wild type [24]. Importantly, while there is extensive evidence to demonstrate expression of ASICs predominantly on peripheral nerve fibers of extrinsic primary afferent neurons from the dorsal root and nodose ganglia, emerging evidence suggests ASIC1–3 expression on other cells including oesophageal epithelial cells [25,26,27]. A study of an in vivo rat model perfused the distal oesophagus with acid solutions through a mucosal perfusion chamber in the presence and absence of ASIC inhibitors [26]. The investigators reported hyperaemia in the presence of CO_2_ challenge, while the administration of generic ASIC inhibitor amiloride was found to inhibit the CO_2_ response, suggesting that luminal perfusion of CO_2_ can diffuse into the oesophageal mucosa, and activate the acid sensors to result in hyperaemia [26].

The nociceptive response of vagal afferent nerves to acidification of the oesophageal mucosa is likely to depend on the activation and downstream sensitization pathway of more than one acid-sensing ion channel [28]. A recent knockout study in mice reported attenuation of acid-induced activation of oesophageal vagal afferent nerves when both ASIC3 and TRPV1 channels underwent knockout simultaneously [29]. Moreover, in cultured human oesophageal epithelial cells exposed to weak acid (pH 5), the potentiating effect of protease-activated receptor-2 (PAR2) agonists trypsin and tryptase were found to induce ATP release through the phosphorylation of both ASIC3 and TRPV1 [30]. This study demonstrated ATP release to be significantly attenuated after treatment with TRPV1 antagonist 5-iodoresiniferatoxin, ASIC3 antagonist amiloride, and PAR2 antagonist, suggesting that PAR2 activation sensitizes human oesophageal epithelial cells to acid through the interdependent activation of ASIC3 and TRPV1 [30].

## 3. Inflammation in Heartburn Pathogenesis

Accumulating evidence has highlighted non-neuronal cells such as immune cells as key mediators of pain pathogenesis and resolution [31]. Most inflammatory mediators can induce acute nociceptive pain by binding their receptors expressed on sensory nerve endings innervating the injured tissue [1]. Conversely, chronic pain has been shown to be regulated by peripheral sensitization of nociceptors, resulting in neuronal plasticity, leading to the notion that the relationship between pain and inflammation is bidirectional. Not only are nociceptive neurons activated by signals released by immune cells, but they can also directly regulate inflammation themselves when induced by noxious stimuli by releasing neuropeptides (inflammatory mediators) such as SP and CGRP which can induce neuro-inflammation [32].

Although the role of inflammation in oesophageal hypersensitivity requires more investigation, recent studies have begun describing mechanisms by which acid exposure may induce an inflammatory state in the oesophageal mucosa [33,34,35,36,37]. A histologic evaluation of rat oesophagus induced ERD surgically by an oesophagoduodenostomy to enable gastric contents to reflux into the oesophagus while maintaining vagal innervation to the stomach. This study demonstrated a T-lymphocyte predominant infiltration of the submucosa at day 3 after surgery, and this infiltration of T cells was found to reach the lamina propria by week 1, and the epithelium by week 3, suggesting a ‘bottom-up’ sequence of mucosal injury that leads to basal cell hyperplasia [34]. These findings were later confirmed in a clinical study which induced acute oesophagitis in patients with previously healed ERD by interrupting their PPI administration, which reinforced the consensus that disease onset follows the infiltration of T lymphocytes into the oesophageal mucosa, followed by the development of hyperplasia [33]. Moreover, patients who developed acute oesophagitis after stopping PPI treatment were found to express increased levels of hypoxia-inducible factor-2α (HIF-2α), a transcription factor that responds to hypoxic stress and regulates inflammation [37,38]. This increase in HIF-2α was coupled with upregulation of proinflammatory cytokines cyclooxygenase-2 (COX2), interleukin-8 (IL8), interleukin-1β (IL1β), tumor necrosis factorα (TNFα), and intercellular adhesion molecule-1 (ICAM-1), coupled with an increase in epithelial cell activity of nuclear factor kappa B (NF-ΚB) and p65 [37]. This is likely due to the translocation of HIF-2α to the nucleus and its stimulation of transcription of target genes via activation of the NF-ΚB-dependent inflammatory pathway in oesophageal epithelial cells exposed to noxious refluxate [39]. These studies collectively highlight a novel hypothesis suggesting that ERD develops from the reflux-induced inflammatory cytokine release by oesophageal epithelial cells which traffic T lymphocyte infiltration into the oesophageal mucosa, thereby causing damage to the oesophageal lining.

A number of clinical studies have further supported an association between pro-inflammatory cytokines and mucosal inflammation in GORD. IL8, a known neutrophil chemoattractant, was overexpressed in the oesophageal mucosa of GORD patients compared to healthy controls in a disease severity-dependent manner [40]. In wild-type mice administered with TRPV1 antagonist capsazepine and acid suppressants, inflammatory parameters were considerably reduced, suggesting that TRPV1 may play a role in acid-induced oesophagitis [41]. Similarly, in a model of rat oesophagitis, SP and isolectin B4-expressing sensory neurons innervating the oesophagus were found to have increased TRPV1 expression in acid-induced oesophagitis [42]. The acid-induced activation of TRPV1 has also been seen on oesophageal epithelial cells in cats, where hydrochloric acid (HCl) challenge was found to induce release of platelet-activating factor (PAF) into the mucosal supernatant [13]. PAF is a chemoattractant for immune cells such as eosinophils which selectively induces their release of reactive oxygen species, thus playing a potential role in oesophageal mucosal damage [43,44]. The exposure of the oesophageal lining in patients with GORD to acid reflux and subsequent activation of TRPV1 may thus act as a catalyst for PAF release and trigger the inflammatory cascade.

However, unlike ERD where mucosal injury is the clear cause for heartburn pathogenesis induced by acid reflux, pain in the absence of macroscopic mucosal injury is more difficult to understand, and increased sensitivity in NERD and FH must occur via different mechanisms. An in vitro study demonstrated the noteworthy role of PAR-2 in pain via regulation of TRPV1 sensitivity. Treatment of cultured oesophageal epithelial cells with the PAR2 agonist trypsin or mast cell tryptase were found to increase ATP release, while this effect was attenuated following treatment of cells with the TRPV1 antagonist 5-iodoresiniferatoxin. These results suggest the possibility that oesophageal epithelial cells are sensitized to acid through TRPV1 phosphorylation via PAR2 activation [30]. The functional importance of PAR-2-mediated pathways in oesophageal hypersensitivity were further highlighted in a study which demonstrated PAR-2 gene upregulation and increased PAR-2 protein expression throughout the ERD and NERD oesophageal epithelium, and a positive correlation with IL8 expression and basal cell hyperplasia [45]. There is also evidence to suggest that activation of TRPV1 in primary afferent nerves causes the release of CGRP and SP when exposed to HCl, thereby causing increased expression of PAF, which can induce inflammation and mucosal damage as a result [13,44]. As such, TRPV1 could be physiologically activated by luminal H+ released during reflux episodes, even in the absence of macroscopic erosions. 

## 4. Microinflammation in GORD

Our understanding of the role of microinflammation in GORD patients without macroscopic lesions (NERD, FH, and hypersensitive oesophagus) remains more limited. While IL8 is known to mediate lymphocyte trafficking in ERD, IL8 mRNA was also found to be upregulated in the oesophageal mucosa of patients with endoscopy-negative GORD compared to healthy controls. An association between high levels of IL8 mRNA, basal hyperplasia, and intraepithelial neutrophils has also been made in NERD patients, suggesting that these smaller numbers of immune cells may be able to induce peripheral sensitization of nerve endings and induce heightened pain sensation in patients with NERD [46,47]. However, immune cell numbers in FH have been demonstrated to be at the same level as asymptomatic subjects, suggesting that microscopic inflammation regulates sensitivity in patients with NERD and hypersensitive oesophagus, but not FH [48]. It is thus possible to suggest that pain in the absence of overt inflammation and persistent heartburn in PPI-refractory GORD could be contributed to by the concept of microinflammation in the oesophageal mucosa.

## 5. Neuro-Immune Interactions in GORD 

Most inflammatory mediators regulate pain by interacting with their receptors located on sensory nerves innervating the injured tissue [1], with chronic pain being maintained by neuronal plasticity via mechanisms of peripheral sensitization [49]. Sensory nerves signal to innate immune cells during the early phases of inflammation, with anatomical studies demonstrating a direct crosstalk between nerve endings and mast cells and dendritic cells [50,51]. Inflammatory mechanisms underlying pain in the absence of overt inflammation have been recently highlighted in functional gastrointestinal disorders such as inflammatory bowel disease (IBS), which has been suggested to overlap with GORD through questionnaire-based diagnoses which showed a positive association between heartburn and IBS symptoms [52]. In these studies, the frequency and severity of abdominal pain was shown to positively correlate with the vicinity of mast cells and colonic mucosal nerves, and this was suggested to result from NGF release by mast cells acting on the NGF receptor NTRK1, expressed on both nerve endings and mast cells, driving neuronal plasticity and persistence of pain [53,54]. This has been shown in animal studies in the colon, where silent visceral afferent nerves were found to be activated by inflammatory mediators released during inflammation such as bradykinin, which led to continuous neuronal firing [55,56].

Neurotrophins such as NGF have been shown to directly upregulate acid-sensing ion channel 3 (ASIC3) gene expression in sensory neurons, suggesting an immunomodulatory role in determining the sensitivity of primary afferent nociceptors [57]. ASICs are activated by protons, and hence can mediate acid perception in the human oesophagus. Our current understanding of the signalling mechanisms of ASIC3 in the oesophageal mucosa remains incomplete, but a study in cultured human oesophageal epithelial cells suggests a likely interaction between PAR-2, TRPV1, and ASIC3 upon its activation [30]. In this study, acid-induced release of ATP, a neurotransmitter frequently involved in pain modulation and inflammation, sensitized oesophageal epithelial cells by phosphorylating TRPV1 and ASIC3 [30]. Moreover, a murine model of NERD highlighted how pharmacological blockade of TRPV1 reduced acid-induced damage to the mucosal integrity of the oesophageal epithelium. In this study, the murine oesophageal mucosa demonstrated an acid-induced reduction in transepithelial electrical resistance (TER), a dynamic measure of mucosal integrity. Administration of TRPV1 antagonist SB366791 and long-term desensitization of the ion channel was found to suppress both the drop in TER, and basal permeability to fluorescein, suggesting a potential role for TRPV1 in mucosal barrier impairment in NERD [58]. A more recent study demonstrated increased ASIC3 expression on oesophageal epithelial cells in patients with ERD and NERD compared to those with FH and BO, suggesting a direct acid-sensing potential of oesophageal epithelial cells in patients with increased noxious stimulation [27].

## 6. Mucosal Neuro-Anatomy

The oesophageal mucosal neural anatomy of patients with different clinical phenotypes of GORD is highly varied. While the proximal oesophagus has been suggested to be more sensitive to chemical, electrical, and mechanical stimuli than its distal counterpart, oesophageal sensation is likely to have multifactorial mechanism of action. The examination of biopsies from the distal and proximal healthy human oesophagus has shown the location of mucosal sensory afferent nerves to be significantly closer to the oesophageal lumen in the proximal oesophagus compared to the distal, as demonstrated by immunoreactivity for both CGRP and PGP9.5 [59]. Recent data suggest that the physical location of nociceptive nerves within the oesophageal mucosa may be able to contribute to peripheral sensitivity in GORD and may even explain why pain can be felt in both NERD and ERD patients. Unlike the deep-lying CGRP-immunoreactive sensory nerves seen in patients with ERD, BO, and FH, patients with NERD have superficial localization of sensory nerve endings which are in very close proximity to the luminal contents of the oesophagus [60]. Thus, NERD patients may experience oesophageal sensitivity at levels similar to patients with erosive lesions due to the ability of ideally located sensory nerves being activated by acidic luminal stimuli, even without significant breach of mucosal barrier integrity. A recent study demonstrated TRPV1 expression on these superficially located sensory nerves in NERD patients, suggesting a directly acid-sensing role for superficial mucosal afferent nerves [27]. In contrast, deeper sensory nerve endings in patients with ERD, BO, and FH did not express TRPV1, and may be more likely to be triggered by mucosal inflammation (which, as described above, is initiated at the deeper layers of the epithelium) [27,33]. These findings highlight TRPV1 as a potential target with therapeutic potential for the treatment of heartburn. While systemic therapy with TRPV1 has proven to be unsuccessful [61], in part due to the difficult balance between efficacy and systemic side effects, topical therapy would be an ideal route of delivery given the superficial localization of TRPV1 on sensory nerves in the oesophageal mucosa of NERD patients.

## 7. The Epithelial Barrier in GORD

The threat imposed by noxious refluxate is first met by the barrier of the squamous epithelium of the oesophagus, which itself provides a protective barrier against luminal contents [62]. Acid-sensing epithelial cells and afferent nerves innervating the oesophageal epithelium induce local homeostatic repair mechanisms [60], and junctional complexes composed of tight junctions, adherens junctions, and desmosomes form a barrier against the diffusion of ions under normal circumstances [61]. However, in patients with ERD where there are clear macroscopic erosions, it is apparent that there is evidence for mucosal barrier deficiency, which is a highly likely candidate for symptom generation. However, in 60% of patients with GORD, macroscopic erosions are not observed. Interestingly, recent studies have highlighted mucosal barrier deficiency even in NERD, where breached mucosal barrier integrity may also contribute to symptom pathogenesis [46]. Electron microscopy studies have identified the presence of dilated intercellular spaces (DIS) in NERD patients, where there is increased distance between oesophageal epithelial cells compared to healthy controls [63]. This morphological measure of impaired epithelial barrier integrity assumes that the increased space between epithelial cells enables noxious luminal content including H^+^ to access and activate nerve endings more readily [64]. Recent studies have suggested that DIS can be induced in response to both acidic and weakly acidic solutions [65], and that it can be resolved by PPI therapy [66].

The morphological change in the oesophageal mucosa was also found to be accompanied by functional changes in the epithelium. The continuous measurement of TER is a functional marker of mucosal integrity, which gives a more dynamic measure of changes in the barrier in response to acid over time, unlike DIS, which is an all-or-nothing phenomenon [67]. In vitro Ussing chamber studies have demonstrated a decreased TER in oesophageal biopsies upon acid exposure, suggesting that patients with heartburn in the absence of apparent mucosal damage have mucosal vulnerability to acid [68]. Furthermore, these studies also pointed out that biopsies from GORD patients had a greater decrease in TER compared to biopsies from control subjects when exposed to acid [68]. Taken together, these studies suggest an inherent mucosal vulnerability in the oesophagus of GORD patients.

Clinical studies investigating functional changes in oesophageal mucosal integrity using intraluminal impedance monitoring have also demonstrated increased mucosal permeability in patients with GORD [69,70]. Oesophageal impedance is an in vivo measure of the dynamic properties of oesophageal mucosal integrity in patients during and after acid challenge. A low impedance measurement suggests that easier ionic passage positively correlates with increased mucosal permeability, and increased sensitivity to perception of acid infusion [70,71]. In patients with NERD, the baseline mucosal impedance was found to be significantly lower than in healthy controls, and impedance values correlated inversely with the severity of 24 h oesophageal acid exposure [69,70]. As such, failure of normal barrier function in NERD may enable passage of noxious components of refluxate to stimulate nerve endings, even in the absence of macroscopic erosions, and hence a vulnerable epithelial barrier may contribute to peripheral sensitization in heartburn pathogenesis [70].

However, noxious refluxate is not the only driver of change in oesophageal epithelial permeability in GORD. Animal studies in rats exposed to acute stress were observed to have DIS and increased oesophageal mucosal permeability to small molecules [72]. Moreover, in mouse colon, exposure to acute stress induced downregulation of tight junction proteins ZO-2 and occludin, while in mouse skin, mast cell degranulation and altered barrier function were reported [73,74]. Mast cells have been similarly highlighted as regulators of stress-induced permeability in rat oesophagus, where there was increased expression of stress response mediator corticotrophin-releasing hormone receptor-2 in response to acute stress [75]. It is important to note that many GORD patients report increased symptom burden with increased stress [76,77], suggesting that the mechanism of heartburn pathogenesis may involve peripheral sensitization driven by changes in oesophageal mucosal permeability.

## 8. Conclusions

The role of central mechanisms of sensitization underlying pathogenesis of heartburn have been well established by earlier studies which infused acid into the oesophagus, measured cortical responses to electrical stimulation, and found central enhancement of sensory transfer in patients with visceral hypersensitivity [78,79]. However, recent studies have shifted focus to investigate mucosal factors that may underlie sensory mechanisms involved in increased sensitivity in GORD patients. These peripheral mechanisms of visceral hypersensitivity may explain the heterogeneity of symptoms observed among patients with similar levels of acid exposure. Acid can activate acid-sensing ion channels expressed on nociceptive neurons and epithelial cells, either directly (as in TRPV1-immunoreactive superficial nerves in NERD) or indirectly via the cytokine-mediated mucosal inflammation, as in ERD. The increased expression of ASIC3 on NERD and ERD patients’ oesophageal mucosa, who have increased acid exposure, compared to FH patients, who do not have increased acid exposure, suggests that ASIC3 expression may be regulated by repeated pathologic acid exposure of the squamous epithelium, and may involve inflammatory pathways. Chemical and inflammatory mediators released during inflammation can heighten the sensitization of peripheral afferent nerve endings and lead to continuous neuronal firing, as seen in animal studies of the colon [55,56]. The adaptive immune response is activated in the onset of ERD, as demonstrated by studies which suggest increased T cell migration induced by acidic bile salts [33,34,80]. This inflammatory response may also regulate the growth and localization of nerve endings, as seen in the colon where mast cells release NGF to induce neuronal sprouting during intestinal inflammation. Disruptions of the oesophageal epithelial barrier functions in ERD and BO due to chronic inflammation can enable repeated exposure of squamous cells and sensory nerves to the noxious refluxate, resulting in a feedback loop of troublesome symptom generation.

This article reviews our current understanding of mucosal sensory mechanisms underlying the generation of reflux-induced symptoms across the GORD spectrum. While much progress has been made, more research is needed to expand our understanding of the role of interdependent interactions between neuronal changes, oesophageal epithelial cells which play a critical barrier and regulatory function, and the inflammatory milieu. These findings will have significant implications for development of topical targets to mitigate heartburn symptoms, including in patients with refractory symptoms.

## Figures and Tables

**Figure 1 ijms-24-02502-f001:**
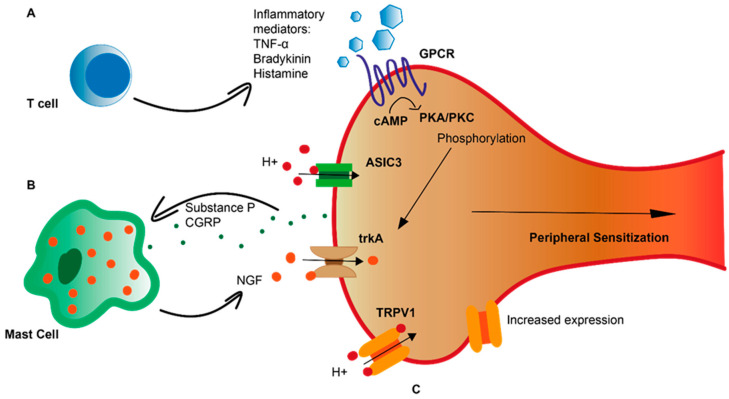
Mechanisms of peripheral sensitization illustration of increased action potential firing in response to a variety of noxious stimuli. (**A**) Mechanisms of nociceptor activation include GPCR activation releasing cAMP and activating protein kinases which phosphorylate ion channels, thus reducing their transduction threshold. (**B**) Upregulation of ion channels such as TRPV1, thereby increasing response to noxious stimuli such as acid. (**C**) Bidirectional neuroimmune crosstalk, including SP and CGRP release from the nerve terminal activating mast cells to release NGF, and NGF activating its receptor trkA expressed on nerve endings. TNF-α: tumor necrosis factor-α; GPCR: G-protein-coupled receptor; cAMP: cyclic adenosine monophosphate; PKA/PKC: protein kinase A/C; ASIC3: acid-sensing ion channel-3; trkA: tropomyosin receptor kinase; TRPV1: transient receptor potential vanilloid-1; CGRP: calcitonin gene-related peptide; NGF: nerve growth factor.

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
