# Peer review of "Sensory Phenotype of the Oesophageal Mucosa in Gastro-Oesophageal Reflux Disease"

_ijms, 2023, doi:10.3390/ijms24032502_

Round 1

Reviewer 1 Report

Manuscript entitled „Sensory Phenotype of the Oesophageal Mucosa in Gastro-Oesophageal Reflux Disease” is very interesting, well-written and well-planned review article. I fully support the publication of this manuscript; however I recommend the minor revision of manuscript. Small corrections should be made to the text according to the following comments:

Introduction

Line 27, 32, 81, 108, 140, 153, 175, 194, 205, 217, 225, 228, 276, 302, 306, 308, 318, 322, 329, 342, 344 – write [2,3]; [4-6]; [12-14]; [23-25]; [31-35]; [35,36]; [42,43]; [11,43]; [45,46]; [49,50]; [52,53]; [54,55]; [61,62]; [69,70]; [70,71]; [69,70]; [73,74]; [76,77]; [78,79]; [54,55]; [31,36,80]

Peripheral sensitisation in GORD

Figure 1 – explain all abbreviations using in the Figure 1 both in the picture and in the description of the figure

Line 74 – write [9,10]

Line 76 – use abbreviation SP instead a full name and remove the full name of abbreviation CGRP. It was explained earlier in the text of manuscript.

Line 80 – explain in full name an abbreviation PGP 9.5

Line 121 - explain in full name an abbreviation PAR2

Inflammation in heartburn pathogenesis

Line 154 – explain all abbreviations in full name: COX-2, IL8, IL1β, TNFα, ICAM-1, NF-ΚB

Line 170 – should be TRPV1

Line 171 - explain in full name an abbreviation HCl

Line 180 - explain in full name an abbreviation FH

Line 183, 187 – write PAR-2

Line 188 – I think should be was not were

Line 193 – use only abbreviation PAF instead of full name

Neuro-immune interactions in GORD

Line 218 - explain in full name an abbreviation GI

Line 235 - write PAR-2

The epithelial barrier in GORD

Line 278 and 279- put the example of reference citation on the end of each sentence

Line 291 - use only abbreviation TER instead of full name. It was explained earlier in text of manuscript

Line 337 – add in patients after mucosa

References

Prepare all references according to rules preferred by the journal as follows:

Journal Articles:

1. Author 1, A.B.; Author 2, C.D. Title of the article. Abbreviated Journal Name Year, Volume, page range.

Another examples of different type of references check in the Instructions for Authors (IJMS)

Author Response

We would like to thank the reviewers for their overwhelmingly positive feedback and constructive comments. Please find an improved manuscript with the changes implemented, and our point by point response below.

Reviewer 1

Introduction:

Line 27, 32, 81, 108, 140, 153, 175, 194, 205, 217, 225, 228, 276, 302, 306, 308, 318, 322, 329, 342, 344 – write [2,3]; [4-6]; [12-14]; [23-25]; [31-35]; [35,36]; [42,43]; [11,43]; [45,46]; [49,50]; [52,53]; [54,55]; [61,62]; [69,70]; [70,71]; [69,70]; [73,74]; [76,77]; [78,79]; [54,55]; [31,36,80]

Response: All in-text citations were changed to the recommended structure.

Peripheral sensitisation in GORD

Figure 1 – explain all abbreviations using in the Figure 1 both in the picture and in the description of the figure

Response: Due to the lack of space in the figure, and the complexity of having full names of ion channels such as transient receptor potential vanilloid-1 within the image, all abbreviations were explained in the Figure caption.

Line 74 – write [9,10]

Response: Change was done

Line 76 – use abbreviation SP instead a full name and remove the full name of abbreviation CGRP. It was explained earlier in the text of manuscript.

Response: Changes were made in the manuscript, line 78-79.

Line 80 – explain in full name an abbreviation PGP 9.5

Response: protein gene product 9.5 written on line 82

Line 121 - explain in full name an abbreviation PAR2

Response: protease-activated receptor 2 written on page 124.

Inflammation in heartburn pathogenesis

Line 154 – explain all abbreviations in full name: COX-2, IL8, IL1β, TNFα, ICAM-1, NF-ΚB

Response: all cytokines have been expanded to include the full name.

Line 170 – should be TRPV1

Response: spelling mistake has been fixed. (LINE 176)

Line 171 - explain in full name an abbreviation HCl

Response: hydrochloric acid was spelt in full.

Line 180 - explain in full name an abbreviation FH

Response: functional heartburn was added to line 34 where the abbreviation is first introduced.

Line 183, 187 – write PAR-2

Response: Change was done.

Line 188 – I think should be was not were

Response: we thank the reviewer for noticing a silly grammar mistake- was has been changes to were.

Line 193 – use only abbreviation PAF instead of full name

Response: abbreviation has been used instead of full name.

Neuro-immune interactions in GORD

Line 218 - explain in full name an abbreviation GI

Response: GI was replaced with gastrointestinal

Line 235 - write PAR-2

Response: change was made.

The epithelial barrier in GORD

Line 278 and 279- put the example of reference citation on the end of each sentence

Response: Citations were moved to the end of each sentence.

Line 291 - use only abbreviation TER instead of full name. It was explained earlier in text of manuscript

Response: full name of TER was replaced with abbreviation. (295)

Line 337 – add in patients after mucosa

Response: ‘patients’ was added to the end of ERD on line 341 for improved grammar.

References

Prepare all references according to rules preferred by the journal as follows:

Journal Articles:

  1. Author 1, A.B.; Author 2, C.D. Title of the article. Abbreviated Journal NameYearVolume, page range.

Response: All references have been updated to the correct format.

Reviewer 2 Report

General comments

This manuscript aimed to review the pathophysiology of peripheral sensitization of esophagus in patients with GERD based on phenotypes, particularly in those with non-erosive oesophageal disease, which constitutes a large patient population with unmet need despite the use of PPIs. The authors did an excellent job to extensively review the mechanistic studies including humans and animals, focusing on different levels of the peripheral sensory pathway.

The major advantages include important data in the last decade such as superficial nociceptors in NERD and cytokine-mediated mucosal inflammation in ERD/BO. The findings may potentiate the possibility of topical or stimuli-specific therapeutic strategies in the future.

There may be some other important points, I think, but were not mentioned in the manuscript:

1.      Secondary hyperalgesia or central sensitization which refers to viscero-visceral or viscero-somatic referred pain due to repeated peripheral stimulation such as esophageal acidification may play an important role in the genesis of reflux-related symptoms including extra-esophageal symptoms.

2.      Stretch sensitive mechanical stimulation provoked by balloon distension or the refluxate, or by longitudinal muscle contraction of esophagus, may also be a possible cause for reflux-related symptoms.

3.      The proximal and distal esophagi may play different role in the generation of reflux symptoms.

4.      Although the esophageal mucosal integrity measured by baseline impedance or DIS seems to be of diagnostic for reflux and possibly pathognomonic for symptom generation, some data showed the co-existence of low baseline impedance and symptom-free. Will you explain this?

5.      There are some typos: the paragraph order number 6 was missing; in the line 170 Should TPV1 be TRPV1?  

Thank you very much for the opportunity to review your excellent paper.   

Author Response

We would like to thank the reviewers for their overwhelmingly positive feedback and constructive comments. Please find an improved manuscript with the changes implemented, and our point by point response below.

There may be some other important points, I think, but were not mentioned in the manuscript:

  1. Secondary hyperalgesia or central sensitization which refers to viscero-visceral or viscero-somatic referred pain due to repeated peripheral stimulation such as esophageal acidification may play an important role in the genesis of reflux-related symptoms including extra-esophageal symptoms.

Response: This is an excellent point and we thank the reviewer for their insight. While the main focus of the paper includes mechanisms of sensitisation happening at the mucosa, a the concept of secondary hyperalgesia has been mentioned in the manuscript on lines 44-47.

  1. Stretch sensitive mechanical stimulation provoked by balloon distension or the refluxate, or by longitudinal muscle contraction of esophagus, may also be a possible cause for reflux-related symptoms.

Response: Stretch sensitive mechanical stimulation has been shown to induce relaxation and contraction of the LES through mechanosensitive neurons which may indirectly induce reflux episodes, but studies measuring the causal relationship of mechanical stimulation to heartburn pathogenesis through mucosal mechanisms was limited, and hence has not been mentioned in the review.    

  1. The proximal and distal esophagi may play different role in the generation of reflux symptoms.

Response: We thank the reviewer for this comment, and have added in a sentence describing neuronal localisation differences between the two oesophageal regions and their possible relevance to reflux generation (Line 260-265).

  1. Although the esophageal mucosal integrity measured by baseline impedance or DIS seems to be of diagnostic for reflux and possibly pathognomonic for symptom generation, some data showed the co-existence of low baseline impedance and symptom-free. Will you explain this?

Response: According to an in vivo study (Woodland, Gut 2012), patients with NERD have a lower baseline impedance and slower post-acid impedance recovery than patients with FH, and the study suggested that increased acid perception is associated with a more vulnerable mucosal integrity maintained by repetitive acid exposures with slower mucosal recovery.  These findings were then confirmed in a more recent study by Xie et al, JNM 2018 where DIS and decreased baseline impedance was only observed in patients with mucosal erosions or pathological reflux, reinforcing the notion that baseline impedance reflects mucosal integrity, and is more sensitive to oesophageal acid exposure.  

  1. There are some typos: the paragraph order number 6 was missing; in the line 170 Should TPV1 be TRPV1?  

Response: This was indeed a typo, and we apologise for the oversight which has now been corrected.